# Comparative Analysis of Long-Term Renal Outcomes in Upper Tract Urothelial Carcinoma: Local Ablation Versus Radical Nephroureterectomy

**DOI:** 10.3390/curroncol32030125

**Published:** 2025-02-22

**Authors:** Blake R. Baer, Meghan V. Matheny, Raidizon H. Mercedes, Jay D. Raman

**Affiliations:** 1Department of Urology, Penn State Health, Milton S. Hershey Medical Center, 500 University Dr., Hershey, PA 17033, USA; bbaer@pennstatehealth.psu.edu (B.R.B.); rmercedes@pennstatehealth.psu.edu (R.H.M.); 2College of Medicine, Penn State College of Medicine, 500 University Dr., Hershey, PA 17033, USA; mmatheny@pennstatehealth.psu.edu

**Keywords:** UTUC, GFR, dialysis, CKD, nephroureterectomy, ablation, urothelial, hematuria

## Abstract

**(1) Background**: Upper tract urothelial carcinoma (UTUC) is typically managed through radical nephroureterectomy (RNU) or local ablation (LA). Compared to RNU, LA offers nephron-sparing benefit for select patients but may present increased recurrence risk. This study primarily compares long-term differences between LA and RNU in chronic kidney disease (CKD) progression, estimated glomerular filtration rate (eGFR) decline, all-cause mortality, and need for dialysis. **(2) Methods**: A retrospective cohort study was conducted using the TriNetX database, examining patients with UTUC treated with RNU (*n* = 2007) or LA (*n* = 4172). Propensity score matching balanced both cohorts (*n* = 1965 per group). Risk ratios and hazard ratios with 95% confidence intervals were calculated over 10 years. **(3) Results**: At 10 years, LA preserved higher mean eGFR (53.49 vs. 46.72; *p* < 0.001) and lower mean creatinine (1.56 vs. 1.66; *p* = 0.017). However, LA held a higher incidence of end-stage renal disease (ESRD) (3.6% vs. 2.2%, *p* = 0.008) and all-cause mortality (26.7% vs. 23.5%, *p* = 0.016). There was no significant difference in rates of dialysis (*p* = 0.79). **(4) Conclusions**: RNU did not carry an increased risk of ESRD, advanced stages of CKD, need for renal dialysis, or overall mortality compared with LA. LA may delay but not totally prevent renal dysfunction when compared to RNU, and exhibits a more gradual timeline.

## 1. Introduction

Upper tract urothelial carcinoma (UTUC) is a rare cancer type comprising 5–10% of urothelial carcinoma and is found in the ureter, intrarenal collecting system, and, most commonly, the renal pelvis. Risk factors for the development of UTUC include advanced age, tobacco use, and history of urothelial cancer of the bladder [1]. The incidence of UTUC has increased over the past three decades and more patients have advanced-stage presentations at diagnosis [2]. The most common presentation is microscopic hematuria with gross hematuria indicating more advanced-stage disease [3]. Ureteroscopy with biopsy is the primary modality for diagnosis and risk stratification of UTUC due to the high false-positive rate from urinary voided cytology alone [4].

UTUC can be surgically treated by local ablation (LA) via retrograde endoscopic or anterograde percutaneous approaches, segmental ureterectomy (SU), or radical nephroureterectomy (RNU). The specific choice of these respective modalities is predicated on disease characteristics or risk level as well as patient-specific factors such as age, performance status, and baseline renal function [5]. The American Urologic Association 2023 UTUC guidelines recommend LA as first-line treatment for favorable low-risk disease, and discuss the possibility of offering it to patients with unfavorable low-risk or favorable high-risk disease who may have focal disease or who may not be suitable candidates for radical surgery [5]. Although RNU is the gold standard for treatment of UTUC (particularly high-risk disease), any associated nephron loss may accelerate progression of chronic kidney disease (CKD) and represents an important limitation of this modality. LA presents an alternative for select high-risk candidates in whom disease parameters are favorable and renal preservation is a top priority. Ideal tumor characteristics for treatment with LA include being low-grade, unifocal, and less than 2 cm and having no invasive features on computed tomography (CT) urography or other axial imaging [6]. Renal preservation is especially impactful to patients with preexisting chronic kidney disease, anatomic solitary kidney, bilateral UTUC, or hereditary cancer syndromes due to increased risk of contralateral upper tract metachronous occurrence [7].

One limitation of LA is risk of local recurrence [8]. Clearly, the remaining upper tract renal unit remains at risk for recurrent disease when compared to those managed by up-front extirpative surgery. However, data suggest that higher local recurrence rates do not necessarily adversely impact cancer survival. Specifically, Grasso et al. showed no difference in cancer-free survival at 2, 5, and 10 years in patients with low-grade UTUC treated with LA versus RNU [9]. Maruyama et al. showed a 90% cancer-specific survival rate at 20 years in patients treated with LA for UTUC [10]. These works have collectively contributed to kidney preservation being an option for low-risk UTUC, as published by guideline panels [5,11].

Despite the promising nature of LA in maintaining oncologic control in patients with UTUC, limited research exists to characterize the long-term impact on glomerular filtration rate decline. Moreover, while RNU has a known risk of causing significant decline in renal function, it is unclear whether LA also carries a risk. The goal of this study, therefore, is to examine a large population database to determine whether there is measurable increased risk of developing CKD, estimated glomerular filtration rate (eGFR) < 15 mL/min/1.73 m^2^, or dependence on dialysis at any recorded point between modalities of treatment for UTUC.

## 2. Materials and Methods

This retrospective cohort review analyzed patients from the TriNetX institutional database research network composed of 104 healthcare organizations and more than 130 million patients using ICD 10 diagnosis codes and CPT treatment codes. Inclusion criteria specified patients aged 18–90 years old with diagnoses of UTUC who were treated by either RNU or LA (endoscopic or percutaneous) between 2004 and 2024 (Appendix A). A propensity-matched analysis was performed between both cohorts based on 38 patient demographics and major comorbidities within 365 days prior to surgical intervention using the TriNetX patient selection algorithm (Appendix B). No crossover between groups was ensured through exclusion of the other treatment group’s CPT codes. Patients within the LA group were excluded from having any type of surgical nephrectomy at any time point to prevent introducing confounding renal loss. TriNetX propensity score matching uses a 1:1 ratio between cohorts using the nearest neighbor method without replacement, with a default caliper of 0.10 standard deviation, and with cohorts randomly shuffled prior to matching (Appendix C). Cohort demographics before and after matching reported in Table 1 and Table 2, along with standard mean difference. After identification and matching, the groups underwent outcome analysis using ICD 10 codes and lab values for 1-, 5-, and 10-year intervals following intervention. For each outcome analyzed, patients who had a diagnosis of that outcome prior to intervention were excluded from that specific analysis. If patient’s data did not extend through the follow-up period, then the last data point in their record was automatically used.

Risk ratios (RRs) and T-tests were calculated for categorical variables and continuous variables, respectively, with *p*-values less than 0.05 representing statistical significance. Kaplan–Meier analysis allowed for detection of hazard ratios (HRs) with 95% confidence intervals (CIs), and proportionality tests, as well as log-rank tests, were used to assess curve differences between cohorts. All statistical analyses were processed using TriNetX built-in analytic software, R v4.0.2, and R Survival package v3.2-3. Graphical data were depicted using GraphPad Prism Version 10.4. The primary outcomes measured were RR and HR of developing CKD, postintervention eGFR, postintervention creatinine level, or risk of requiring dialysis within 10 years after treatment between modalities of treatment for UTUC.

## 3. Results

A total of 4172 patients in the LA group and 2007 patients in the RNU group met the initial inclusion criteria. From these larger cohorts, 1965 patients from each group from across 60 institutions were included in the matched analysis. Prior to propensity matching, patients who underwent LA at baseline were older, a greater proportion were male, a greater proportion were Asian, and they had higher rates of renal tubule–interstitial disease, obstructive and reflux uropathy, urolithiasis, unspecified urinary tract infections (UTIs), cystitis, and stage 4 CKD. They also had greater rates of other comorbidities such as diabetes, metabolic disease, cerebrovascular disease, hematuria, and urinary retention. The LA group had lower values for preoperative baseline creatinine, other disorders of kidney and ureter, rates of ESRD, contracted kidneys, dialysis, and glomerular diseases (Table 1).

Following intervention, mean creatinine (1.56 ± 1.21 for LA and 1.66 ± 1.18 for RNU, *p* = 0.017) and mean eGFR (53.49 ± 25.61 for LA and 46.72 ± 18.65 for RNU, *p* < 0.001) were significantly worse for RNU at 10 years. When assessing the cumulative risk of developing ESRD, stage 5 CKD, requiring dialysis, or having eGFR <15 mL/min/1.73 m^2^, 185 (10.0%) LA patients and 144 (7.8%) RNU patients were identified, with the LA group having a slightly increased risk (RR 1.29, 95% CI 1.05–1.59, *p* = 0.017) and higher hazard ratio at 10 years (HR 1.27, CI 1.02–1.58). Their respective curves are significantly different between cohorts on log-rank test (*p* = 0.029) (Figure 1).

At 10 years postintervention, there was no significant difference between the LA group (*n* = 58, 3.0%) and RNU group (*n* = 55, 2.8%) regarding those in need of dialysis postintervention, with an RR of 1.05 (*p* = 0.79) and HR of 1.037 (CI 0.72–1.50). However, 70 (3.6%) LA patients and 42 (2.2%) RNU patients had a new diagnosis of ESRD at 10 years postintervention (RR 1.65, CI 1.13–2.41, *p* = 0.008).

By the conclusion of the 10-year follow-up period, 377 (21.3%) LA patients and 376 (21.2%) RNU patients developed new stage 4 CKD or eGFR <30 mL/min/1.73 m^2^ (RR 1.005, CI 0.89–1.14, *p* = 0.93), with an HR of 0.947 (CI 0.82–1.09). However, the curves demonstrated a significant difference between groups on log-rank analysis (*p* = 0.0024) and with significant non-proportionality of the HR (χ^2^ = 33.305, *p* < 0.001), implying variance in diagnosis rates over time. This significant difference is demonstrated between matched groups at 1 year, with 220 (12.4%) LA patients and 272 (15.3%) RNU patients being diagnosed by the end of 1 year (RR 0.81, CI 0.69–0.96, *p* = 0.013) (Figure 2).

Rates of new stage 4 CKD, stage 5 CKD, or eGFR <30 mL/min/1.73 m^2^ were similar between groups, with 393 (22.9%) LA patients and 373 (21.5%) RNU patients being diagnosed at the end of 10 years (RR 1.06, CI 0.94–1.21, *p* = 0.33), and at 1 and 5 years for matched or unmatched cohorts.

When comparing cumulative 10-year risk and rate of developing new diagnoses of any stage of CKD or ESRD, no significance was found between groups, with 480 (31.7%) LA patients and 426 (30.2%) RNU patients being diagnosed (RR 1.05, CI 0.94–1.17, *p* = 0.37) (HR 1.018 CI 0.89–1.16). Matched and unmatched groups were not significantly different at 1, 5, or 10 years, and cohorts demonstrated comparable progression curves on the log-rank test (χ^2^ = 0.070, *p* = 0.791). The same findings were true with new diagnosis of only advanced stages of CKD (stage 3 CKD or greater) which did not significantly differ between groups at 10 years following intervention (RR 1.05, *p* = 0.462), or at 1, 5, or 10 years in matched or unmatched cohorts (Figure 3), with the log-rank test demonstrating similar curves between cohorts (χ^2^ = 0.115, *p* = 0.734).

The groups demonstrated a disparate timeline in median days to the development of diagnosis of CKD and associated eGFR: stage 3 CKD (LA 764 days vs. RNU 110 days), stage 4 CKD (LA 828 days vs. RNU 427 days), stage 5 CKD (LA 2021 days vs. RNU 1085 days), stage 3 CKD or greater (LA 537 days vs. RNU 25 days), stage 4 or 5 CKD (LA 1398 vs. RNU 464 days) (Table 3).

The overall risk of 10-year all-cause mortality between groups was significantly different, with 524 (26.7%) LA patients and 458 (23.5%) RNU patients dying (RR 1.14, CI 1.03–1.27) but with an insignificant HR of 1.11 (CI 0.98–1.26).

Within 10 years following intervention, 45.9% of LA patients and 37.0% of RNU patients received some form of antineoplastic therapy (RR 1.24, CI 1.15–1.34).

## 4. Discussion

In our study, renal functional outcomes following UTUC intervention were compared between propensity-matched cohorts from TriNetX.

Before propensity matching, the LA group was qualitatively more comorbid, having more urologic disorders, non-urologic disorders, and increased age compared to RNU group. Interestingly, we found the RNU group had worse preexisting renal function, evidenced by increased baseline creatinine and higher preintervention rates of ESRD and dialysis, with the exception that the LA group, however, had higher baseline rates of stage 4 CKD. The higher preoperative creatinine and rates of ESRD or dialysis within the RNU group was different from what we would have anticipated. While a vast portion of treatment modality is dictated by tumor biology and risk stratification, guidelines recommend offering LA as an initial management strategy in select patients with even high-risk disease who may be at risk of renal failure [5]. One explanation for our findings is a delayed ICD timeline such that a diagnosis code appeared in a patient’s record before the CPT code for treatment. Alternatively, providers may not deem it necessary to preserve renal function in patients who already have ESRD or require dialysis and opt for better oncologic control at the expense of greater nephron loss. Because underlying comorbidities, such as diabetes, age, or underlying renal function, predispose individuals to an increased risk of developing significant chronic renal insufficiency [12], they may and should factor into the choice between treatments.

The TriNetX propensity matching effectively reduced a significant portion of differences between cohorts, based on SDM reduction. Of note, the presence of preoperative antineoplastic therapy within 1 year prior to intervention was measured but not included in the matching process. After matching, the cohorts remained different for this variable with 11.2% of LA patients and 16.1% of RNU patients having received antineoplastic therapy within 1 year prior to intervention. Propensity matching may reduce some differences between groups, but patient and provider decision-making also factors in comorbidities, operative complication rates [13], consideration for chemotherapy [14], genetic predisposition to metachronous contralateral UTUC [5], invasiveness of treatment, oncologic outcomes [13,15], family history of cancer, and life expectancy, among others.

When considering treatment modality for UTUC, patients and their providers are encouraged to take renal function into account [5], especially assessing risk of postoperative CKD. Previous studies have shown significant decreases in eGFR following RNU [3,12,16,17,18,19]. Our study found the mean reduction of eGFR in the LA group to be −7.01 mL/min/1.73 m^2^ (11.5%) and in the RNU group to be −13.06 mL/min/1.73 m^2^ (21.8%) across 10 years. This finding coincides well with prior studies by Labbate et al. in scope (18–32%) of RNU nephron loss and with similar postoperative eGFR of 40.8 mL/min/1.73 m^2^ compared to our study of 46.7 mL/min/1.73 m^2^ [17,18]. That same study speculated that almost 80% of patients undergoing RNU without neoadjuvant therapy would experience new diagnosis of stage 3 CKD or greater [17], though other studies have suggested lower percentages [20,21]. Our study was discordant with this, finding only 24.4% of patients received a formal diagnosis stage 3 CKD or greater within 10 years post-RNU, without consideration of neoadjuvant systemic therapy. However, our study excluded 348 patients with prior stage 3 CKD or greater, so total risk of stage 3 CKD or greater would be closer to 45.9%. Some studies have speculated some transiency in the loss of renal function following RNU, which may also account for our study’s less-than-expected CKD progression rate [3,22]. The significant comorbidity of the LA group, along with older age, may factor into these patients opting for more renal-preserving therapies, regardless of their UTUC risk level.

To our knowledge, this study is the first to analyze comparative risk of dialysis following intervention for UTUC between LA and RNU. We found overall no significant difference between groups at 10 years (*p* = 0.79), even in the unmatched cohort (*p* = 0.78). Previous studies have shown preservation of renal function after LA, with one study showing 94% of patients retaining normal preoperative renal function [9]. Similarly to our cohort’s results, other studies also demonstrated a reduced decline in eGFR for LA patients compared to RNU (−8.24 mL/min/1.73 m^2^ versus −13.37 mL/min/1.73 m^2^, respectively, *p* = 0.032) [15,18]. The overall eGFR loss reported by our study was 11.5% for LA and 21.8% for RNU, which indicates a clear reduction in renal function following treatment. There is benefit to exploring renal-preserving therapies to assess impact on renal function and subsequent prevention of clinical outcomes such as the need for dialysis and the risk of further morbidity or mortality.

The primary goal of this study was not to interrogate oncologic outcomes and their relationship with renal function; however, this topic is essential in the conversation about LA compared to RNU. Prior studies have demonstrated that eGFR does not have a significant alteration on cancer-specific outcomes after RNU [16], but that it may have a significant role in prognosis [23,24], both in overall survival [24] or cancer-specific survival [25]. This raises questions about the interplay of renal function and oncologic outcomes. While our study is limited by its inability to discern UTUC grade, risk-stratify patients, follow recurrence-free survival, or clarify cancer-specific mortality, it does show all-cause mortality to be higher in the LA group at 10 years, even after matching (RR 1.143, *p* = 0.016). In addition, we found that local ablation across 10 years demonstrated higher rates of new ESRD (*p* = 0.008), though RNU resulted in significantly worse eGFR and creatinine values. We did not find a significant difference between both groups in risk of requiring dialysis, new diagnosis of CKD (any stage), or acute kidney injury (AKI) frequency.

Renal-preserving therapy with LA is an appealing choice for patients trying to avoid CKD or need for dialysis, but our study found that the rate of requiring dialysis within 10 years was not any higher in one group compared to the other (*p* = 0.79). In addition, our study showed that there was no difference between groups in diagnosis of all stages of CKD (*p* = 0.37), including advanced stage 3 CKD or greater (*p* = 0.46). When we evaluated the combined risk of stage 5 CKD, GFR < 15 mL/min/1.73 m^2^, ESRD, or dialysis, we found an overall increased risk at the end of 10 years for LA (RR 1.28, *p* = 0.017). This may perhaps be attributable to the LA cohort’s worse underlying comorbidities and not to the treatment modality itself.

Of particular interest, our study did not find a significant difference in risk of progression to stage 5 CKD (*p* = 0.158), although it did find significance in those progressing to ESRD (*p* = 0.008). Only 3.6% of LA and 2.2% of RNU patients in our study progressed to ESRD compared to in a study by Chen et al. which found an insignificant difference in rates of 38% in LA and 34% in RNU, respectively [18]. The proportion of patients is vastly different; however, the greater quantity of LA patients progressing to ESRD compared to RNU is consistent. A study by Abrate et al. found this trend to be true for SU compared to RNU as well, indicating that intervention types may not significantly influence progression to ESRD. They showed there to be largely insignificant differences between patients postoperatively regarding eGFR loss or 5-year overall survival following SU vs. RNU [26]. These authors concluded that eGFR should not be so much considered as an outcome but as a condition affecting the outcome.

We judged the relationship with chemotherapy to be a large distractor from the primary endpoint of our study, and this was purposefully excluded from our narrowed examination scope. However, neoadjuvant and adjuvant chemotherapy plays a crucial role in the calculus for modality of treatment preference and is itself closely linked to renal function and outcomes [5,15]. Expert guidelines recommend offering neoadjuvant chemotherapy in instances where poor postoperative renal function would prohibit platinum-based chemotherapy [5]. For instance, elderly patients and patients with pre-RNU eGFR closer to 60 mL/min/1.73 m^2^ are more likely to be ineligible for adjuvant cisplatin-based chemotherapy regimens because of renal function loss after RNU [14], and only 55% of patients are candidates for chemotherapy after RNU if the cutoff for eGFR >45 mL/min/1.73 m^2^ [27]. Our study is intrinsically biased by not controlling for chemotherapy as a confounder between both groups. In our matched cohort, 11.2% of LA patients and 16.1% of RNU patients received antineoplastic therapy within 1 year prior to intervention, representing a significant difference between cohorts. Within 10 years following intervention, 45.9% of LA patients and 37.0% of RNU patients received some form of antineoplastic therapy (RR 1.24, CI 1.15–1.34). However, this difference between groups should be considered in terms of its confounding effect on renal function and for further exploration in future studies.

One of the notable findings from our study was the overall differing timelines of renal loss. Our study found a sudden and acute decline in kidney function, particularly within the first week, for patients who underwent RNU. Within the first week postintervention, 6% of RNU patients and 1% of LA patients were diagnosed with an AKI. Almost half (44%) of RNU patients versus only 7% of LA patients who would go on to develop stage 4 CKD or greater were diagnosed within the first week. This trend extended to almost all data points, which may mostly be expected given the abrupt surgical nephron loss from RNU. This was also evidenced by the difference in median days to diagnosis for almost all primary data points, with RNU usually demonstrating a faster timeline to diagnosis (Table 3). In addition, Figure 2 visually demonstrates the difference between cohort curves, with non-proportional HR indicating variance in the rates of diagnoses. However, renal function loss is still present but not as rapid for patients who undergo LA. It has been pointedly observed previously that nephron-sparing surgery like LA may delay but not totally prevent the development of CKD [28]. Dudinec et al. performed a retrospective cohort study between nephron-sparing surgery such as LA or SU and RNU and found a cumulative incidence of advanced CKD in 17.8% of LA or SU patients vs. 29.9% of RNU (*p* = 0.009) and at 10 years, these values rose to 24.6% and 39.9%, respectively. Furthermore, this study found an initial decline in eGFR following RNU but a more gradual decline in the nephron-sparing approaches, and determined that the two curves met at around 4–5 years after surgery [28]. This coincided with a key point from our study, which demonstrated a significant reduction in renal function for patients with RNU, but a more gradual reduction in LA over time such that the two groups were mostly insignificantly different at 10 years (Figure 2). The differences between the groups in renal preservation seem to diminish with time, even with attempts at matching to reduce comorbidities as confounders.

While this study has several valuable insights, its applicability is limited. It is a retrospective, non-randomizable cohort study that relies on complete and accurate ICD 10 codes and CPT codes for identification of patients, their outcomes, and their timelines. Loss to follow-up or incomplete recording may weaken data interpretation. Additionally, there are limitations by TriNetX to identify valuable oncologic-specific variables such as cancer-specific mortality, pathologic T-stage, urine cytology, recurrence-free survival, UTUC risk level, or histologic grade, which would be helpful to qualify our findings. Our study underestimates impact on renal function by excluding patients with outcomes prior to intervention. Our study also does not extensively examine the involvement of chemotherapy on renal function. The greater proportion of preoperative RNU patients or postoperative LA patients who receive antineoplastic therapy may have an unequal risk for renal dysfunction. A final noteworthy limitation is that propensity matching attempts to remove confounding variables which might influence outcomes, but this process is imperfect and limited. There are real differences between groups that are unable to be entirely reduced. As well, propensity matching may limit this study’s external validity by falsely negating many of these real differences between groups, which may be unhelpful for clinicians and patients alike in interpreting its results.

In summary, the choice between renal-preserving therapy such as LA and RNU should consider eGFR and impact on renal function but should primarily be based on surgical and oncologic risk factors [26]. We found there to be no significant increased risk of dialysis, stage 5 CKD, stage 4 CKD, stage 3 CKD, or new diagnosis of any-stage CKD, between either intervention across 10 years. LA carried increased risk for all-cause mortality, and new development of ESRD at 10 years. It also held an increased combined risk of new stage 5 CKD, GFR < 15 mL/min/1.73 m^2^, ESRD, or dialysis at the end of 10 years. RNU had overall worse renal function measured by creatinine and eGFR. The timeline of our study demonstrates that there is a real preservation of renal function for patients who underwent LA as compared to RNU in the short term; however, the observable difference is diminished over a 10-year window of follow-up. While head-to-head prospective studies examining rates of dialysis or transplantation would be near-impossible in this field, further exploration is warranted regarding impacts of these treatments on the relationship with the patient’s cancer biology, need for adjuvant chemotherapy, impact on quality of life, and identifying other contributors to the development of advanced stages of CKD.

## 5. Conclusions

In this large, propensity-matched, retrospective cohort series, we found that RNU did not carry an increased risk of ESRD, advanced stages of CKD, need for renal dialysis, or overall mortality compared with nephron-sparing LA. Patients who underwent LA were overall more likely to develop ESRD with higher all-cause mortality at 10 years, although the RNU group had lower mean eGFR and mean creatinine postintervention. Matched analysis may obviate some but not all confounding comorbidities between groups, and renal dysfunction may be intrinsically linked with the treatment timeline of each modality. LA may delay but not totally prevent renal dysfunction when compared to RNU, and exhibits a more gradual timeline. When counseling patients about treatment possibilities, clinicians should account for impact on renal function but should continue to prioritize surgical and oncologic outcomes.

## Figures and Tables

**Figure 1 curroncol-32-00125-f001:**
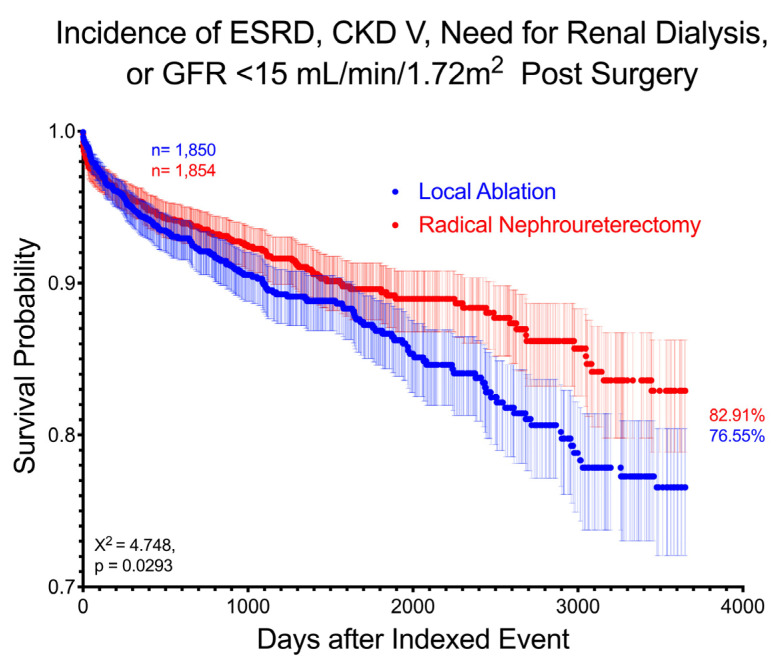
Survival curve demonstrating difference between LA and RNU in incidence of new ESRD, stage 5 CKD, dependence on renal dialysis, or eGFR <15 mL/min/1.72 m^2^ over 10 years after intervention. Patients with prior diagnoses are excluded. Log-rank test χ^2^ = 4.748, *p* = 0.0293.

**Figure 2 curroncol-32-00125-f002:**
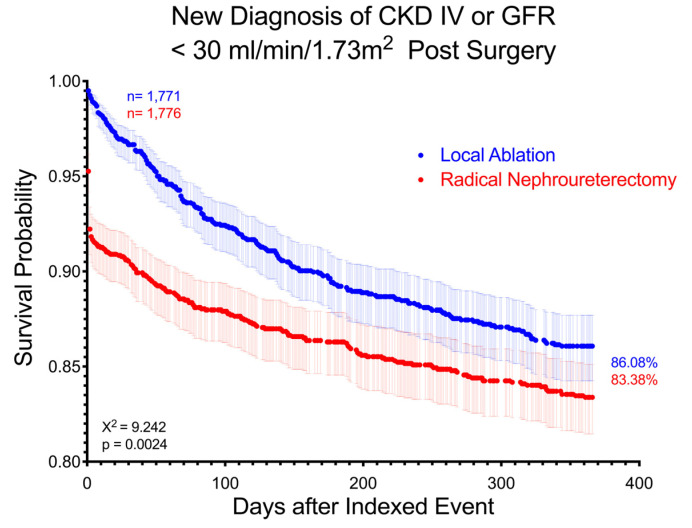
Survival curve demonstrating group difference between LA and RNU in new diagnosis of stage 4 CKD or eGFR <30 mL/min/1.73 m^2^ over 1 year after intervention. Patients with prior diagnoses are excluded. Log-rank test χ^2^ = 9.242, *p* = 0.0024.

**Figure 3 curroncol-32-00125-f003:**
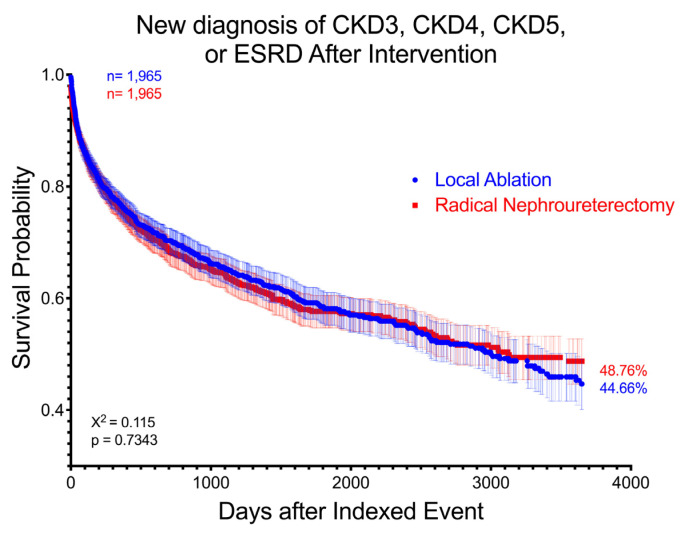
Survival curve demonstrating absence of group difference between LA and RNU in new diagnosis of stage 3 CKD, stage 4 CKD, stage 5 CKD, and ESRD over 10 years after intervention. Patients with prior diagnoses are excluded. Log-rank test χ^2^ = 0.115, *p* = 0.7343.

**Table 1 curroncol-32-00125-t001:** Patient demographics before propensity matching cohorts, with SMD between groups. Patients were propensity scored according to comorbidities identified in the preceding 365 days prior to but not including the day of intervention.

		Before Matching		
		LA	RNU	
		N = 4172	N = 2007	Standard Mean Diff.
Gender	Male	2670 (64.0%)	1224 (61.0%)	0.062
	Female	1414 (33.9%)	724 (36.1%)	0.047
	Unknown	88 (2.1%)	59 (2.9%)	0.051
Age at Index		71.0 ± 9.6	69.2 ± 10.0	0.182
Race	White	3179 (76.2%)	1487 (74.1%)	0.049
	Black or African American	213 (5.1%)	82 (4.1%)	0.049
	Asian	222 (5.3%)	54 (2.7%)	0.134
	American Indian or Alaska Native	10 (0.2%)	10 (0.5%)	0.043
	Native Hawaiian or Other Pacific Islander	12 (0.3%)	0 (0.0%)	0.076
	Other	139 (3.3%)	59 (2.9%)	0.022
	Unknown	397 (9.5%)	315 (15.7%)	0.221
BMI [kg/m^2^]		28.2 ± 6.0	28.2 ± 6.2	0.001
Creatinine [Mass/volume]		1.2 ± 0.7	1.3 ± 0.9	0.063
GFR [mL/min 1.73 m^2^]		60.8 ± 22.8	59.5 ± 20.4	0.060
AKI and CKD N17–N19		924 (22.1%)	421 (21.0%)	0.028
AKI N17		481 (11.5%)	204 (10.2%)	0.044
CKD N18		710 (17.0%)	339 (16.9%)	0.003
	CKD III N18.3	417 (10.0%)	183 (9.1%)	0.030
	CKD IV N18.4	121 (2.9%)	41 (2.0%)	0.055
	CKD V N18.5	14 (0.3%)	10 (0.5%)	0.025
	ESRD N18.6	37 (0.9%)	40 (2.0%)	0.093
	Dialysis CPT 1012740	10 (0.2%)	14 (0.7%)	0.067
Glomerular Diseases N00–N08		45 (1.1%)	37 (1.9%)	0.064
Renal Tubulo-Interstitial Diseases N10–N16		1539 (36.9%)	645 (32.1%)	0.100
Acute Pyelonephritis N10		42 (1.0%)	22 (1.1%)	0.009
Obstructive and Reflux Uropathy N13		1500 (36.0%)	619 (30.8%)	0.109
Urolithiasis N20–N23		616 (14.8%)	239 (11.9%)	0.084
Other Disorder of Kidney and Ureter N25–N29		1675 (40.1%)	1053 (52.5%)	0.249
Unspecified Contracted Kidney N26		60 (1.4%)	71 (3.5%)	0.135
Other Diseases of Urinary System N30–N39		1461 (35.0%)	563 (28.1%)	0.150
Cystitis N30		427 (10.2%)	123 (6.1%)	0.150
UTI, Site Unspecified N39.0		733 (17.6%)	293 (14.6%)	0.081
Essential Hypertension I10		1917 (45.9%)	879 (43.8%)	0.043
Ischemic Heart Disease I20–I25		855 (20.5%)	369 (18.4%)	0.053
Cerebrovascular Disease I60–I69		267 (6.4%)	89 (4.4%)	0.087
Diabetes E08–E13		843 (20.2%)	336 (16.7%)	0.089
Metabolic Disorders E70–E88		1801 (43.4%)	786 (39.2%)	0.086
Chronic Lower Respiratory Diseases J40–J4A		718 (17.2%)	306 (15.2%)	0.053
Alcohol-related Disorders F10		71 (1.7%)	28 (1.4%)	0.025
Nicotine Dependence F17.2		295 (7.1%)	141 (7.0%)	0.002
Hematuria R31		1933 (46.3%)	820 (40.9%)	0.111
Retention of Urine R33		260 (6.2%)	81 (4.0%)	0.100
Proteinuria R80		96 (2.3%)	35 (1.7%)	0.040

**Table 2 curroncol-32-00125-t002:** Patient demographics after propensity matching cohorts, with SMD between groups. Patients were propensity scored according to comorbidities identified in the preceding 365 days prior to but not including the day of intervention.

		After Matching		
		LA	RNU	
		N = 1965	N = 1965	Standard Mean Diff.
Gender	Male	1202 (61.2%)	1202 (61.2%)	<0.001
	Female	713 (36.3%)	705 (35.9%)	0.009
	Unknown	50 (2.5%)	58 (2.9%)	0.027
Age at Index		69.2 ± 10.0	69.3 ± 9.9	0.012
Race	White	1454 (74.0%)	1464 (74.5%)	0.012
	Black or African American	93 (4.7%)	81 (4.1%)	0.030
	Asian	52 (2.6%)	54 (2.7%)	0.006
	American Indian or Alaska Native	10 (0.5%)	10 (0.5%)	<0.001
	Native Hawaiian or Other Pacific Islander	0 (0.0%)	0 (0.0%)	-
	Other	61 (3.1%)	58 (3.0%)	0.009
	Unknown	295 (15.0%)	298 (15.2%)	0.013
BMI [kg/m^2^]		28.6 ± 6.1	28.2 ± 6.2	0.065
Creatinine [Mass/volume]		1.2 ± 0.7	1.3 ± 0.8	0.044
GFR [mL/min 1.73 m^2^]		60.5 ± 21.2	59.8 ± 20.3	0.034
AKI and CKD N17-N19		424 (21.6%)	407 (20.7%)	0.021
AKI N17		195 (9.9%)	203 (10.3%)	0.013
CKD N18		332 (16.9%)	325 (16.5%)	0.010
	CKD III N18.3	186 (9.5%)	179 (9.1%)	0.012
	CKD IV N18.4	37 (1.9%)	40 (2.0%)	0.011
	CKD V N18.5	10 (0.5%)	10 (0.5%)	<0.001
	ESRD N18.6	27 (1.4%)	31 (1.6%)	0.017
	Dialysis CPT 1012740	10 (0.5%)	10 (0.5%)	<0.001
Glomerular Diseases N00–N08		33 (1.7%)	33 (1.7%)	0.010
Renal Tubulo-Interstitial Diseases N10–N16		628 (32.0%)	636 (32.4%)	0.009
Acute Pyelonephritis N10		18 (0.9%)	22 (1.1%)	0.020
Obstructive and Reflux Uropathy N13		640 (31.0%)	611 (31.1%)	0.001
Urolithiasis N20–N23		233 (11.9%)	237 (12.1%)	0.006
Other Disorder of Kidney and Ureter N25–N29		1012 (51.5%)	1016 (51.7%)	0.004
Unspecified Contracted Kidney N26		39 (2.0%)	69 (3.5%)	0.093
Other Diseases of Urinary System N30–N39		594 (30.2%)	554 (28.2%)	0.036
Cystitis N30		122 (6.2%)	123 (6.3%)	0.004
UTI, Site Unspecified N39.0		286 (14.6%)	289 (14.7%)	0.003
Essential Hypertension I10		843 (42.9%)	859 (43.7%)	0.045
Ischemic Heart Disease I20–I25		365 (18.6%)	358 (18.2%)	0.009
Cerebrovascular Disease I60–I69		92 (4.7%)	88 (4.5%)	0.010
Diabetes E08–E13		331 (16.8%)	329 (16.7%)	0.003
Metabolic disorders E70–E88		752 (38.3%)	767 (39.0%)	0.016
Chronic Lower Respiratory Diseases J40–J4A		302 (15.4%)	300 (15.3%)	0.003
Alcohol-related Disorders F10		32 (1.6%)	28 (1.4%)	0.017
Nicotine Dependence F17.2		142 (7.2%)	136 (6.9%)	0.012
Hematuria R31		799 (40.7%)	806 (41.0%)	0.007
Retention of Urine R33		73 (3.7%)	81 (4.1%)	0.021
Proteinuria R80		32 (1.6%)	35 (1.8%)	0.012

**Table 3 curroncol-32-00125-t003:** Comparison of median days to diagnosis of each stage of CKD and associated eGFR over 10 years after LA and RNU, demonstrating disparate timelines between cohorts.

CKD (eGFR mL/min/1.73 m^2^)	LA	RNU	HR (CI)
Stage 3 or greater (0–59)	537 days	25 days	0.58 (0.51–0.66)
Stage 3 (30–59)	764 days	110 days	0.58 (0.51–0.65)
Stage 4 (15–29)	828 days	427 days	0.95 (0.82–1.09)
Stage 5 (0–14)	2021 days	1085 days	1.13 (0.91–1.41)

## Data Availability

The data presented in this study are available through TriNetX LLC (https://trinetx.com/), although an account may be required for access. Date of access: 12 December 2024. TriNetX Engineering (2023). *TrinetX*. R package version 0.1.0.

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
