# Peer review of "Comparative Analysis of Long-Term Renal Outcomes in Upper Tract Urothelial Carcinoma: Local Ablation Versus Radical Nephroureterectomy"

_curroncol, 2025, doi:10.3390/curroncol32030125_

Round 1
Reviewer 1 Report
Comments and Suggestions for Authors
The authors utilized a large retrospective data set to compare renal function outcomes for patients undergoing renal preservation with local ablative therapies to nephroureterectomy for upper tract urothelial carcinoma. The authors adequately acknowledged the limitations of study methods.
1. The conclusions in the abstract are not supported by the presented results. The conclusions presented in the body of the manuscript are appropriate.
2. These findings highlight the need to prioritize oncologic principles when treating patients with upper tract urothelial carcinoma.
Reviewer 2 Report
Comments and Suggestions for Authors
I read with great interest the manuscript "Comparative Analysis of Long-Term Renal Outcomes in Upper Tract Urothelial Carcinoma: Local Ablation Versus Radical Nephroureterectomy" by Baer et al., which investigates the renal outcomes after local ablation or RNU.
Abstract
The indication of LA and RNU are different in UTUC. Authors should mention that the RNU is the standard first-line treatment for high-risk patients, while LA is, in most cases, optional for patients with low-risk UTUC.
Method
1. baseline testing in observational/registry studies is not important, there are numerous reasons for this, but the main one is that the analysis should reduce confounding. P-values set arbitrary threshold for assessing a different in groups were know are different. A p-value indicating "statistical significance" for a baseline characteristic does not provide meaningful insight into whether the characteristic is clinically or practically important. Conversely, non-significant results do not confirm equivalence between groups, and can falsely reassure. Report by group for context (as is done) but remove the p-values from table 1 and the methods describing this.
European Urology's guidance for the presentation of statistics, and reporting observational research is available freely at:
https://doi.org/10.1016/j.eururo.2018.12.014 IF: 25.3
2. propensity score matching: Provide the adjusted mean difference (standard mean difference) instead of p values. Provide the matching ration (here is 1:1 as the table 1 showed), caliper value, matching method (“nearest”?)
If possible, provide the code of propensity score matching in TriNetX built-in analytic software, I am not familiar with this software, maybe the author can try ‘MatchIt’ package in R software.
Result
1. I strongly recommend that you sperate Table 1 into two tables, one as Patient demographics before matching without p value (also delete the p value in the Results), another as Patient demographics after matching with adjusted mean difference (standard mean difference).
2. Why were there missing data after matching. PSM can only be done for complete data cases, which means if a parameter was involved in PSM, a patient with missing data in this parameter will be excluded automatically. For example, after matching, 1202 female + 661 male = 1863 in LA group (n = 1965), around 100 cases should be excluded. Please ask for help from a professional statistician.
3. For all figures, the text and figure did not match. In line 122 ‘These differences were insignif-122 icant at 1- and 5- years, and with 10-year overall HR of 1.27 (CI 1.02-1.58, p=0.10)(Figure 1)’. Again, please learning how to use “survival” and “survminer” package in R to build a good survival figure.
4. For median days (figure 4), a table would be better to present the descriptive statistics indicators. delete Figure 4.
Overall, I will only reconsider providing a complete review comment if the authors improve the tables and figures as I have suggested. I think the conclusion is not reliable as I am not sure how the PSM was performed.
Reviewer 3 Report
Comments and Suggestions for Authors
The authors analyzed the long-term renal outcomes of a retrospective cohort of patients with upper tract urothelial carcinoma (UTUC) under local ablation (LA) and radical nephroureterectomy (RUN) treatment.
Some comments are listed below.
1. Line 15: Provide the full name of CKD since it is the first time it is mentioned.
2. Lines 43 and 49: chronic kidney disease CKD
3. Lines 74 and 77: Appendix 1 A and Appendix 2 B
4. Table 1: The LA group (N= 4208) number doesn't match the total number of males (2693)+ females (1424). The same issue applies to the RUN group. Please recheck all the numbers in Table 1.
5. Figure 1: p = 0.0293, but in the figure legend, the authors described that "Survival curve demonstrating absence of difference between local ablation and radical nephroureterectomy in incidence of ESRD, CKD stage 5, dependence on renal dialysis or eGFR <15" mL/min/1.72m2 over 10 years after intervention. Log rank test χ2 = 4.748, p = 0.0293.”. Please explain or correct this conclusion.
6. Figure 3: p = 0.7343, but in the figure legend, the authors described that "Survival curve demonstrating group difference between local ablation and radical nephroureterectomy in new diagnosis of CKD stage 3, CKD stage 4, CKD stage 5, ESRD or eGFR <30 ml/min/1.73m2 over 10 years after intervention. Log rank test χ2 = 0.115, p = 0.7343.” Please explain or correct this conclusion.
7. Figure 4: Provide the p-value or (*) mark directly in the figure to visualize when comparing the LA and RUN groups.
8. Lines 182-184: The authors described that "Interestingly, we found the RNU group had worse preexisting renal function evidenced from significantly increased baseline creatinine and higher pre-intervention rates of ESRD and dialysis.” Could the authors discuss the guidelines or criteria of treatment options for LA and RUN? Are the contents that the authors mentioned due to the process of treatment (LA or RUN) selection?
Reviewer 4 Report
Comments and Suggestions for Authors The article is about the comparison of renal function preservation between local treatment and total nephroureterectomy for upper urothelial carcinoma. It is a significant study with a large number of cases.  Since local treatment is being considered, the reviewer assumes that low-risk lesions as defined by the EAU guideline are the target of the study. What was the T stage of the lesion and the urine cytology positivity rate? The authors also mentioned that there was no difference in the percentage of patients who went on dialysis, but the reviewer thinks it is also important to know the percentage of local recurrence or distant metastasis of the lesions and the percentage of cancer deaths. Additionally, the reviewer has the impression that the discussion is quite long.Author Response
Please see the attachment.

Round 2
Reviewer 2 Report
Comments and Suggestions for Authors
Authors should be congratulated for their work. The revised version the manuscript answered to all my concerns, improving the quality of the manuscript.
The manuscript is now suitable for publication.
Reviewer 3 Report
Comments and Suggestions for Authors
Thanks for the revision.
Comments on the Quality of English LanguageN/A